# Learning Efficient Parameter Server Synchronization Policies for Distributed SGD

Rong Zhu*, Sheng Yang, Andreas Pfadler, Zhengping Qian, Jingren Zhou
Alibaba Group
* Corresponding Author

## Abstract

We apply a reinforcement learning (RL) based approach to learning optimal synchronization policies used for Parameter Server-based distributed training of machine learning models with Stochastic Gradient Descent (SGD). Utilizing a formal synchronization policy description in the PS-setting, we are able to derive a suitable and compact description of states and actions, allowing us to efficiently use the standard off-the-shelf deep Q-learning algorithm. As a result, we are able to learn synchronization policies which generalize to different cluster environments, different training datasets and small model variations and (most importantly) lead to considerable decreases in training time when compared to standard policies such as bulk synchronous parallel (BSP), asynchronous parallel (ASP), or stale synchronous parallel (SSP). To support our claims we present extensive numerical results obtained from experiments performed in simulated cluster environments. In our experiments training time is reduced by $44\%$ on average and learned policies generalize to multiple unseen circumstances.

## 1 Introduction

In recent years, Stochastic gradient descent (SGD) Bottou (2012) and its variants Kingma & Ba (2014); Chen et al. (2016), have been adopted as the main work horse for training machine learning (ML) models. To be able to train large models, which are both computationally demanding or require very large training datasets, SGD is often parallelized across several machines, with the well-known *parameter-server (PS)* framework being one of the most widely adopted distribution strategies. In the PS setting, there commonly exist one (or several) parameter servers and multiple worker nodes. The parameter server maintains the globally shared model parameters and aggregate updates from workers. Each worker node pulls the latest model parameters from the server, computes all gradients and pushes them back for updating. As this approach generally reduces the amount of inter-node communication, it may provide for considerably reduced training time.

**Challenges of Distributed SGD in PS.** In the PS setting, a central task is to design a **synchronization policy**, which coordinates the execution progress of all workers. This synchronization policy determines in each step, i.e. whenever a gradient is pushed by some worker, the state ("run" or "wait") of each worker, until the next update arrives at the parameter server. Thus, it directly determines the overall training time. However, finding a good synchronization policy is difficult, as this will at least depend on the properties of the underlying optimization problem and the nature of the cluster used for training.

We briefly review some existing policies and discuss their limitations: In the simple bulk synchronous parallel (BSP) policy Valiant (1990), the parameter server waits for all workers to push their updated gradients, and then lets them pull the same latest model parameters for the next step. However, in a heterogeneous cluster, it is common that some workers, also referred to as *stragglers*, run much slower than others. Waiting for these straggler workers certainly decreases the number of SGD iterations per unit of time and leads less optimal usage of available computational resources.

The asynchronous parallel (ASP) policy alleviates the straggling problem by allowing each worker to run immediately after it pushes its gradients. Therefore, fast workers can move ahead without waiting for others. However, worker may push *stale* gradients that are evaluated on an older version of the model parameters, which may have a negative impact on the overall convergence speed Chen et al. (2016); Cui et al. (2016). Thus, neither BSP nor ASP consistently outperform the other on different models and datasets Zinkevich et al. (2009); Dutta et al. (2018).

A better policy is, for instance, given by the stale synchronous parallel (SSP) policy Ho et al. (2013); Dai et al. (2015), which can be considered as a trade off between BSP and AP. For SSP one defines a fixed threshold $s$, such that the fastest worker is allowed to outpace the slowest one by at most $s$ steps ("bounded staleness"). However, an optimal value of $s$ is hard to specify without enough prior knowledge. Moreover, the optimal value of $s$ may change during the entirety of the training process. Other synchronization policies (see in Section 2) may either degenerate to the above mentioned three policies or may need extensive manual tuning of hyper-parameters. Now, the key problem addressed in this work is the following: *how can we design a synchronization policy to **automatically** and* ***adaptively*** *optimize the overall training time of SGD in PS?* Note that in the following, whenever we speak of "overall training time" or "time until convergence", we specifically mean the amount of time needed until a model reaches a certain pre-defined accuracy on a validation dataset.

**Our Key Contributions.** To tackle this problem, we focus on designing automatic synchronization policies for distributed SGD in a PS setting. Instead of relying on expert experience we try to *learn* a better synchronization policy using a reinforcement learning (RL) approach with training data being obtained by observing the execution process of PS-based distributed SGD.

To this end, we first represent the synchronization problem in the distributed SGD training in a unified framework, allowing us to formally describe both existing (BSP, ASP and SSP) and learned policies. Based on this framework, we formulate the synchronization policy design problem as an RL problem, where we reward those policies requiring less training time. To train an RL-based synchronization policy (RLP), we carefully design the state and action space of the RL model, such that it is able to generalize to different training data, models and cluster environments, while still ensuring efficient policy learning process.

Our model is trained using the off-the-shelf deep Q-Learning algorithm Mnih et al. (2013); Van Hasselt et al. (2016). Furthermore, we design a pre-training process to speed up the convergence. Empirical results demonstrate the validity of our approach and the advantages of our learned RLP policy in terms of training efficiency and generalization ability. In our experiments, RLP improves overall training time by $44\%$ on average in comparison to the best existing policy. More over, RLP is able to generalize to multiple unseen circumstances.

**Limitations.** We note the following limitations of this work: First, our experiments are based on "plain" SGD in its simplest form, e.g. without momentum or other adjustments. While our framework may in principle be used with any first-order optimization scheme, we have chosen to refrain from using more commonly used methods, such as Adam Kingma & Ba (2014) in order to reduce the number of hyper-parameters and allow for a clearer differentiation between different policies.

Second, we note that all of our experiments were performed in a simulated cluster environment, allowing us to easily change the number of workers used and artificially create training instances characterized by different levels of straggling workers, resp. gradient staleness.

Third, due to resource constraints, we have not applied our approach to the training of very demanding models, such as deep Convolutional Neural Networks Krizhevsky et al. (2012) or transformer-based models like BERT Devlin et al. (2018), which are now considered the "state of the art" in Computer Vision and NLP, respectively. We leave experiments on this scale for future work.

Despite the above limitations, we argue that our results clearly indicates the applicability of our approach in practice. In our experiments, we were not only able to demonstrate the mere existence of optimal synchronization policies for an individual pair of underlying model and training data, but also show that learned policies are able to provide significant speedups, even when applied to training slightly different models on different training datasets and different cluster environments.

## 2 RELATED WORK

PS-based training with distributed SGD can be considered a standard method for training large ML models. Finding proper synchronization policies coordinating all workers to reduce training time has been a long-standing problem. For example, the classic BSP policy Valiant (1990) has been directly applied for distributed SGD in a PS-setting. As the performance of BSP is heavily diminished by the straggling problem, other variants, such as the $K$-sync BSP and $K$-batch-sync BSP have been proposed in Dutta et al. (2018). They alleviate the straggling problem by slightly relaxing the

synchronization condition. However, as the global synchronization barrier still exists, workers tend to spend a considerable amount of time in an idle state. Other ways to speed up the BSP policy include reassigning data Harlap et al. (2016) and adding backup workers Chen et al. (2016). However, they may case additional, non-negligible communication overhead Jiang et al. (2017). Another extreme is to run distributed SGD using the ASP policy, which avoids the straggling problem but typically forces one to consider the staleness problem. The negative effects of staleness have been shown in Zinkevich et al. (2009); Chen et al. (2016); Cui et al. (2016); Dutta et al. (2018); Dai et al. (2019) on typical ML models, where Dai et al. (2019) performed a comprehensive experimental analysis. To correct bias caused by staleness, Hadjis et al. (2016); Mitliagkas et al. (2016); Zheng et al. (2017) proposed some methods to compensate for delayed gradients or tune the momentum parameter, which also increases the overall computational costs.

To combine the advantages of BSP and ASP together, Ho et al. (2013); Dai et al. (2015) proposed the SSP policy as a trade-off. SSP may often lead to faster convergence when compared to both BSP and ASP. However, the threshold $s$ in SSP which bounds the staleness is hard to tune and fixed during the training process. To overcome this, Jiang et al. (2017); Zhao et al. (2019) proposed a dynamic SSP policy where $s$ is tuned during the training process. Fan et al. (2018) proposed a more flexible adaptive asynchronous parallel policy to allow different values of $s$ for each worker. Although more adaptive, they need manual tuning of hyper-parameters. As a result, until now, there appears to exist no synchronization policy that is both fully adaptive and automatic.

This paper, for the best of our knowledge, marks the first instance where RL is used find synchronization policy for distributed SGD in a completely data-driven fashion. RL has been successful applied to control robotics Duan et al. (2016) and games Mnih et al. (2013); Silver et al. (2016). Recently, it has been widely used to optimize problems such as task scheduling Mao et al. (2019), resource management Mao et al. (2016) and optimization Li & Malik (2016); Marcus et al. (2019). Here, we formulate the search for efficient synchronization policies as an RL problem and derive RL-based policies (RLP) which can be applied to training a specific underlying model (and variations thereof): Once such a policy has been learned, it can be reapplied to future training instances. This is particularly relevant in situations where the same model (or type of model) is trained regularly on different training data sets on the same cluster of machines (or similar cluster in the case of e.g. dynamically allocated cloud computing resources).

*Overall, to the best of our knowledge, our proposed RLP provides for the first time synchronization policies which are adaptive, automatic and avoid both the straggling and staleness problems.*

## 3 METHOD

We describe the technical details of our proposed method in this section. First of all, we present a unified framework to represent the synchronization problem of distributed SGD in the PS-setting in Section 3.1. Based on this, we formalize the synchronization policy design problem as an RL problem in Section 3.2. Section 3.3 discusses how to train the RL-based policy.

### 3.1 A UNIFIED FRAMEWORK

We introduce a unified framework generalizing all existing policies (BSP, ASP and SSP) and providing us convenience and flexibility to design new synchronization policies. The pseudo-code of the framework is shown in Algorithm 1.

In a PS environment, let $S$ be the parameter server holding the global parameter $\omega$ and $\mathcal{W} = \{W_1, W_2, \ldots, W_k\}$ be a set of workers. For each worker $W_i$, we regard its computation process as a series of *steps*. In each step, the state of worker $W_i$ is either *active* or *idle*. An active worker pulls the latest parameter $\omega$ from $S$, performs the SGD computation on a mini-batch of data and pushes the gradient $\nabla\omega$ back to $S$. Then, $W_i$ is scheduled by $S$ to run or wait. On the server side, $S$ iteratively receives gradients sent by workers and coordinates their execution progress until the converge condition is met. $S$ maintains the global parameter $\omega$ and a set $I$ of all idle workers. Initially, we set $I = \emptyset$. To avoid ambiguity, we refer to the period of $S$ receiving two consecutive gradients as an *iteration*. In each iteration, $S$ receives a gradient $\nabla\omega$ submitted by a worker $W_s$ and updates the global parameter $\omega$ according to the SGD update rule. After that, $W_s$ is added into the idle set $I$, and $S$ selects a subset of active workers $A \subseteq I$ according to the synchronization policy. All workers

---

**Algorithm 1: Unified Synchronization Policy Framework**
**Input:** a server $S$ and a set of workers $\mathcal{W} = W_1, W_2, \ldots, W_k$
**Output:** trained global parameter $\omega$ and total time cost $t$

1: $I \leftarrow \emptyset, t \; gets 0$
2: initialize the global parameter $\omega$
3: set all workers in $\mathcal{W}$ to be active
4: **while** stopping condition of SGD is not met **do**
5:      $S$ receives gradient $\nabla\omega$ submitted by worker $W_s$
6:      update parameter $\omega$ to $\omega'$ by the SGD rule
7:      $I \leftarrow I \cup \{W_s\}$
8:      select set $A \subseteq I$ according to the synchronization policy
9:      record time cost $\Delta t$ and set $t \leftarrow t + \Delta t$
10: **end while**
11: **return** $\omega$ and $t$

---

in $A$ are allowed to run while all other workers in $I - A$ keep idle. Let $\Delta t$ denote the iteration time cost. The global parameter $\omega$ and total time $t = \sum \Delta t$ is returned after convergence (i.e. when a certain stopping criterion is met).

By specifying a different set $A$ of active workers in each iteration, our framework can describe various synchronization policies. For example:

• BSP: we set $A = \emptyset$ if $|I| < k$ and $A = I$ otherwise, so all workers execute together only when all of them finish a computation step;
• ASP: we set $A = \{W_s\}$, so that all workers always continue running after submitting the gradients;
• SSP: let $p_i$ denote the number of computation steps performed by worker $W_i$. Given a threshold $s$, for the submitted worker $W_s$, if $0 < p_s - \min_j p_j < s$, we set $A = \{W_s\}$; otherwise if $p_s - \min_j p_j = s$, we set $A = \emptyset$; otherwise when $p_s = \min_j p_j$, we set $A = I$, so all workers outpace the slowest one by no more than $s$ steps.

In each iteration, there exist at most $2^{|\mathcal{W}|}$ different choices. Finding a better policy using some combinatorial search algorithms or hand-crafted heuristic rules seems intractable due to the large search size. Thus, we aim at designing a method to automatically learn a synchronization policy.

## 3.2 FORMULATION AS AN RL PROBLEM

Based on the unified framework, we can formulate the synchronization policy design problem as an RL problem. To lay foundation, we briefly review some preliminaries of RL. The general setting of RL is shown in Fig. 1, where an agent continuously interacts with an environment. In each step $n$, the agent observes some state $S_n$ of the environment and is asked to take an action $a_n$. Following this action, the environment emits a reward $r_n$ and transitions to state $S_{n+1}$. The state transitions and rewards are both stochastic and satisfies the Markov property, i.e. the state transitions and rewards depend only on the state $S_n$ and action $a_n$. The goal of the agent is to learn a sequence of actions chosen by observing the states to maximize the expected cumulative reward $\mathbb{E}[\sum_n r_n]$.

Notice that, the agent picks actions based on a policy $\pi$, which is a probability distribution over state-action pairs: $\pi(S, a) \rightarrow [0, 1]$. In most practical problems, it is impossible to store the policy in tabular form but more common to represent it as a function $\pi_\theta$ parametrized by $\theta$. Recently, deep neural networks (DNN) have been widely used to represent $\pi_\theta$ in many RL problems Mnih et al. (2013); Li & Malik (2016); Van Hasselt et al. (2016); Marcus et al. (2019); Mao et al. (2019). Following this trend, in our work, we also adopt a DNN to represent the RL policy. Feeding it with a state vector $S$, it outputs a value $\pi(S, a)$ for all possible actions $a$.

We find that the synchronization policy design problem resembles a prototypical RL problem, as it aims at learning how to choose an active worker set (action) in each iteration based on the SGD execution progress (state) to optimize the total time cost (reward). Fig. 1 illustrates how to set the three key components in the RL formulation. The details are elaborated as follows:

• *State*: we choose features in each SGD iteration to characterize its execution progress. For generalization purposes, the state feature vector should be able to represent clusters with different number of workers. To this end, we regard the execution of all workers as a black-box and just encode the information they submitted in the server side. For each iteration $n$, we just record the feature tuple

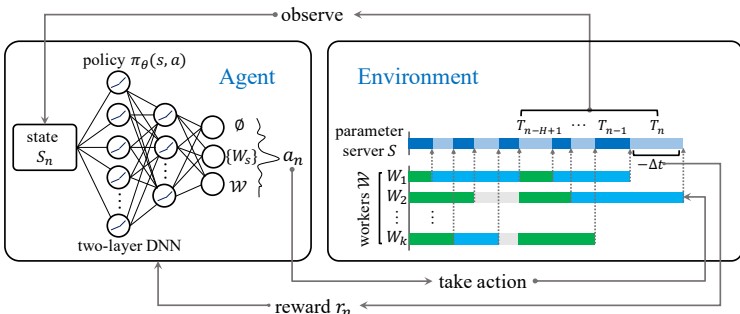

Figure 1: Illustration of formulating the synchronization policy problem as an RL problem.

$T_n = (n, L(\omega), L(\omega') - L(\omega), \ell_n)$, where $L(\omega)$ and $L(\omega')$ are the loss values before and after one iteration. $\ell_n$ records the number of pushed gradients by other workers during the execution step of this submitted worker, which reflects the level of staleness of the submitted gradients. Notice that, $T_n$ is irrelevant to the number of workers in the cluster. Meanwhile, we do not use the gradient information in our features since it is highly dependent on the underlying ML models and would thus impede generalization. The state vector $S_n = (T_n, T_{n-1}, \ldots, T_{n-H+1})$ keeps track of the features of the $H$ most recent iterations. At the very beginning, we pad the historical information all by zero.

• **Action**: the largest action space contains at most $2^{|\mathcal{W}|}$ possible actions, which is relevant to the worker number and impossible to train for large clusters. In our setting, we choose a very small but still powerful action space. Let the action $a_n \in \{\emptyset, \{W_s\}, \mathcal{W}\}$. That means we have three valid actions in each iteration: $\emptyset$ keeps all idle worker to be idle; $\{W_s\}$ allows the submitted worker itself to run; and $\{\mathcal{W}\}$ allows all idle workers to run. Setting such a small size action space enables faster training of the RL policy. As we have shown earlier in Section 3.1, this action space is enough to represent and switch between the existing policies BSP, ASP and SSP with different threshold $s$. Moreover, it can also represent more complex policies.

• **Reward**: we directly set the reward signal in each iteration as $r_n = -\Delta t$. Hence, maximizing the cumulative reward corresponds to finding policies minimizing the total time cost.

### 3.3 TRAINING RL POLICY

It has been shown in Gu et al. (2016) that the off-policy RL algorithms such as Q-learning can be more sample efficient than their policy gradient counterparts. This is largely due to the fact that policy gradient methods require on-policy samples for the new policy obtained after each update of the policy parameters. Therefore, we adopt the standard deep Q-learning method Mnih et al. (2013); Van Hasselt et al. (2016) to perform an end-to-end training of our RL policy. Our policy network is a two-layer neural networks with 64 and 32 units in each hidden layer, respectively. Leaky rectified activation units are used in the two hidden layers. We present the detailed training process in Algorithm 2.

The RL policy training process is embedded into the SGD algorithm. Each training episode corresponds to training the underlying ML models once by SGD. To stabilize the learned policy, we apply an evaluation policy network $Q$ and a targeted policy network $Q^*$ with parameters $\theta$ and $\theta^*$, respectively. The parameters of $Q$ are copied to $Q^*$ every $c$ iterations. We maintain an experience replay pool $\mathcal{D}$ with size $N$ to store transitions $(S', a, r, S)$, where $S'$ is the previous state of $S$. In each iteration, we sample a mini-batch $\mathcal{B}$ of transitions from $\mathcal{D}$. For each transition $(S_{i-1}, a_{i-1}, r_{i-1}, S_i)$, let $y_i = Q(S_{i-1}, a_{i-1}; \theta)$ and $\hat{y}_i = r_{i-1} + \gamma \max_a Q(S_i, a; \theta)$ denote the estimated and targeted cumulative reward of the current state with discount factor $\gamma \in (0, 1]$. We apply a square loss $\|y_i - \hat{y}_i\|_2$ to train the parameters $\theta$ of $Q$. After that, we choose the action $a$ in an $\epsilon$-greedy manner in order to decide the execution status of workers. Finally, the policy network $Q^*$ is returned for inference.

In our synchronization policy design problem, we observe that interval of possible training times of SGD under different policies has large overlaps due to the stochastic nature of SGD. To help the policy network to distinguish the difference of synchronization policies earlier and speed up the converge, we integrate the policy network with some prior knowledge on existing synchronization policies. Specifically, we apply a pre-training process to train the policy network in advance. We execute SGD with existing policies (BSP, ASP and SSP with different thresholds $s$) and record the information state $S$ and action $a$ for each iteration. After SGD finishes, we obtain the truly

**Algorithm 2: Training RL Policy**

1: initialize the experience replay pool $\mathcal{D}$ with size $N$
2: **for** episode $\leftarrow 1$ to $T$ **do**
3:     **while** the stopping condition of SGD is not met **do**
4:         obtain the current state vector $S$ and reward $r \leftarrow -\Delta t$
5:         store the transition $(S', a, r, S)$ into the replay pool $\mathcal{D}$
6:         sample a mini-batch $\mathcal{B}$ of transitions from $\mathcal{D}$
7:         train the parameters $\theta$ of $Q$ by the squared loss of $y_i$ and $\hat{y}_i$ on $\mathcal{B}$
8:         copy the parameters $\theta$ of $Q$ to $\theta^*$ of $Q^*$ every $c$ iterations
9:         choose action $a \leftarrow \begin{cases} \text{random action with probability } \epsilon \\ \arg\max_a Q^*(S, a; \theta^*) \text{ with probability} 1 - \epsilon \end{cases}$
10:       apply action $a_t$ to set the execution status of workers
11:       $S' \leftarrow S$
12:     **end while**
13: **end for**
14: **return** policy network $Q^*$ with parameters $\theta^*$

cumulative reward $-t$, i.e. the training time left until the end, for each iteration. Then, we apply the squared loss on the difference of $Q(S, a; \theta)$ and $-t$ over all iterations to train the parameters of the policy network. We observed that using pre-training policy exploration time may be saved and faster convergence achieved.

## 4 EXPERIMENTS

We implement RLP in a simulated cluster/PS environment. This allows for convenient creation of various training instances. We now report our evaluation results in this section.

**Instance Generation Method.** We train the RLP by training a DNN model consisting of several fully connected layers on a simulated cluster of 10 workers. In the following we refer to this model as the "underlying" model (as opposed to the RL policy model).

For each training instance, we need to configure both the cluster and the underlying model. To simulate stragglers, we have a probability $p = 0.3$ to activate the *sleep()* function in each worker in the cluster. The sleeping time obeys a Gaussian distribution. For the underlying DNN model, we randomly choose a number $h \in \{0, 1, 2, 3\}$ as the number of hidden layers. When $h = 0$, DNN degenerates to a multi-class logistic regression model. Each hidden layer contains $256$ units with rectified activation functions. We use cross-entropy as the loss function. In each instance, we randomly sample $50\%$ data from the MNIST dataset and run the standard SGD for training.

**Training and Testing Methods.** The hyper-parameters for RLP are set as follows: historical size $H = 10$, replay pool size $N = 50$, mini-batch size $|\mathcal{B}| = 32$, copy rate $c = 5$, discount factor $\gamma = 0.8$, exploration probability $\epsilon = 0.1$ and learning rate to be $0.01$. For the underlying DNN model, we set its batch size to 16 and learning rate to $0.01$. SGD terminates once we attain $88\%$ validation accuracy. Before training, we apply the existing policies BSP, ASP, and SSP with $s = 2, 5, 8$ to pre-train the policy network on 100 instances, respectively. Then, we train the RLP policy until convergence with about 1,000 instances (episodes). When testing we run each instance 30 times with different random seeds under the same policy and report the average time cost.

**Performance Comparison vs. Existing Synchronization Policies.** First of all, we examine the performance of RLP by comparing its execution time with respect to BSP, ASP, and SSP. The result of SSP refers to the result for the best threshold $s$. We report the average results tested on 100 instances in Fig. 2(a). On the whole, our proposed RLP runs $1.56$, $1.86$ and $1.44$ times faster than BSP, ASP and SSP, respectively. This verifies that our RL-based method can find better synchronization policies in the case of our particular choice of underlying model.

To present more details, we also compare the average testing results on different models and different cluster environments. Fig. 2(b) reports the results for the underlying model with 0 and 3 hidden layers. RLP improves the running time by $48\%$ and $43\%$ w.r.t. the best existing policy, respectively. There exists no significant difference on the speedup ratio of RLP w.r.t. different models. This is most likely due to very similar loss curves for these model variations. Therefore, the relative difference of different policies tends to be similar. Fig. 2(c) reports the results on clusters with

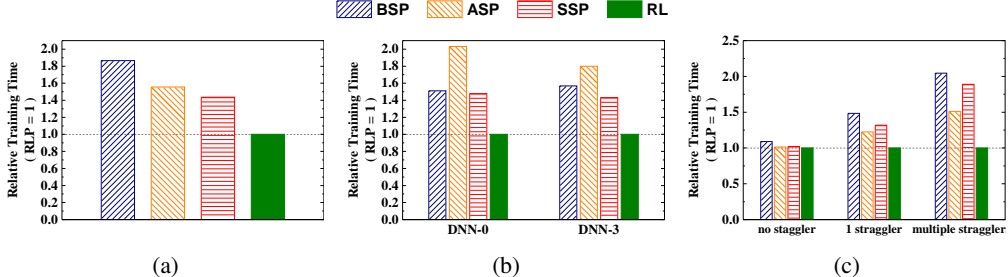

Figure 2: Performance comparison of RLP w.r.t. existing policies. (a) Average results on all testing cases. (b) Results on DNN models with 0 and 3 hidden layers. (c) Results on clusters with different number of stragglers.

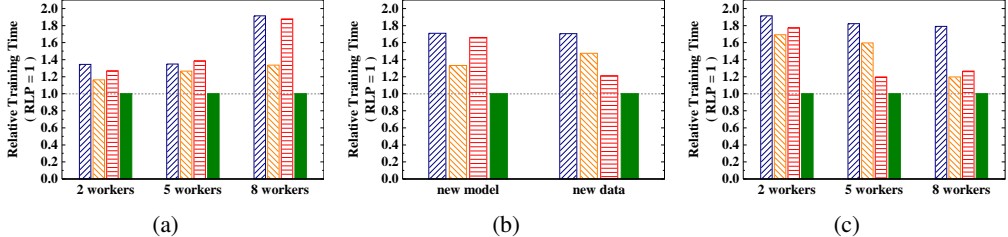

Figure 3: Evaluation on the generalization ability of RLP on unseen cases. (a) Results on clusters with different number of workers. (b) Results on new DNN model and new dataset. (c) Results on combination of clusters with different number of workers, new DNN model and new dataset.

different number of stragglers. RLP improves the time cost by $2\%$, $22\%$ and $51\%$ w.r.t. the best existing policy when having $0$, $1$ and multiple stragglers, respectively. The speedup ratio is more obvious when having more stragglers. This is due to when there exists no stragglers, all existing policies perform in a similar way, so there leaves no room for RLP to improve anymore. By our observation, RLP performs in the same way as the optimal ASP this time. When there exist more stragglers, the straggling and staleness problems of all workers become more complex, so there exists more space for RLP to explore new better policies. These detailed evaluation shows that RLP is both efficient and adaptive to speed up distributed SGD in different circumstances.

**Generalization Ability of RLP.** In this set of experiments, we evaluate the generalization ability of RLP by applying it to process unseen instances with different configuration of clusters and models. First, we consider the generalization to clusters with different number of workers. We randomly choose $2, 5$ and $8$ workers in the cluster and apply RLP to train the underlying DNN model having one hidden layer. The results are illustrated in Fig. 3(a). We observe that RLP policy trained on a cluster having 10 workers also performs much better than the existing policies on cluster with different number of workers. We argue that this is due to the fact that in our RL formulation, both the state and action representation are irrelevant to the number of workers. Therefore, our RLP policy is easy to transfer to a different cluster environment. Second, we examine the generalization ability of RLP w.r.t. new model and new data and show result in Fig. 3(b). For a previously unseen DNN model with 6 hidden layers and a new CIFAR10 dataset, RLP improves the training time by $33\%$ and $22\%$ w.r.t. the best existing policy, respectively. This shows that our trained RLP can also generalize to train similar underlying ML models with unseen training datasets. This could be due to the fact that we record only the information of the loss value in the state representation of RLP. Thus, training models with similar loss curve may also speed up by our RLP policy experience. Third, we combined these testing cases together to evaluate RLP on a new DNN model and dataset with a different number of workers. Fig. 3(c) shows that RLP also achieves good performance in this setting. It improves the running time by $69\%$, $19\%$ and $20\%$ w.r.t. the best existing policy on $2$, $5$ and $8$ workers, respectively. In summary, this set of experimental results shows that our design of the states and actions for RLP should allow it to generalize to different settings.

**Detailed Insights into RLP.** Finally, we look more int the details of the learned RLP policy, To this end, in Fig. 4, we visualize the execution process of all workers in a typical case with multiple stragglers, where the color and gray blocks represent that the worker is running and idle, respectively. Tab. 1 summarizes statistics for each policy. Based on this, advantages and disadvantages of each policy become visible and we obtain the following insights:

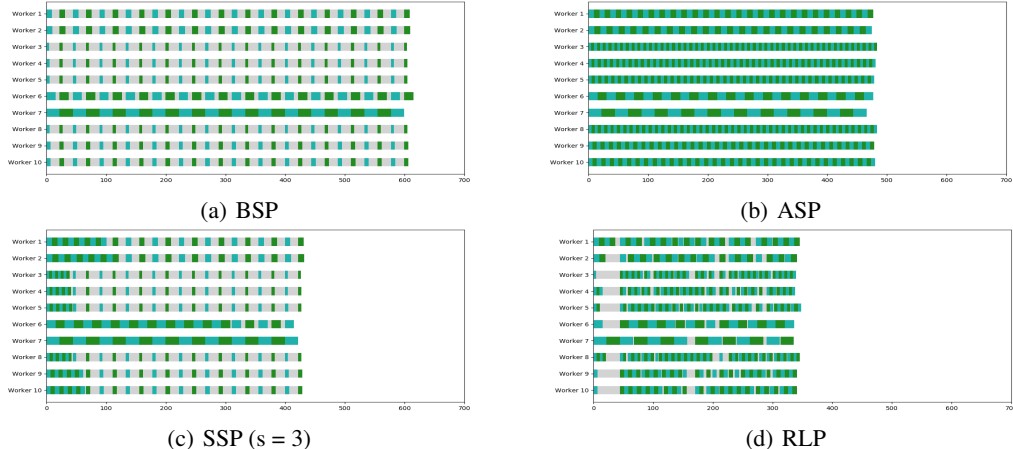

(a) BSP

(b) ASP

(c) SSP (s = 3)

(d) RLP

Figure 4: Visualization of the training process of SGD under different synchronization policies.

Table 1: Statistics for different policies as observed in our experiments.

| Policy | $W_1$ | $W_2$ | $W_3$ | $W_4$ | $W_5$ | $W_6$ | $W_7$ | $W_8$ | $W_9$ | $W_{10}$ | # iterations | % idle time | % inference time |
|---|---|---|---|---|---|---|---|---|---|---|---|---|---|
| BSP | 28 | 28 | 28 | 28 | 28 | 28 | 28 | 28 | 28 | 28 | 279 | 54.24% | 0 |
| ASP | 52 | 47 | 100 | 93 | 87 | 31 | 21 | 91 | 70 | 67 | 569 | 0 | 0 |
| SSP | 26 | 26 | 26 | 26 | 26 | 25 | 19 | 26 | 26 | 26 | 252 | 43.5% | 0 |
| RLP | 34 | 28 | 53 | 51 | 51 | 19 | 14 | 54 | 38 | 38 | 380 | **8.1%** | **1.4%** |
| Step Time | 9.17 | 10.103 | 4.37 | 4.47 | 4.57 | 15.39 | 22.189 | 5.31 | 4.97 | 5.07 | — | — | — |

- Both BSP and SSP force all workers to do almost the same number of computation steps but spend around half of the time in an idle state. In fact, SSP degenerates to BSP in the later steps. Therefore, although BSP performs less iterations than ASP, its total training time is longer.
- For ASP, there exists no synchronization barrier for workers to wait, so the number of steps done by workers are highly correlated to their step time (the correlation coefficient is around $-0.9$ ). However, due to the increased staleness level, ASP performs a larger number of iterations (2.4 times than BSP).
- For RLP, we have two observations. First, RLP only synchronizes workers less times than necessary. We may observe very short idle times (only $8\%$) for RLP. Thus, RLP appears to never block the fastest workers. Similar to ASP, the number of steps done by workers is also highly correlated to their step time. Second, the synchronization barriers chosen by RLP are all worthwhile. We find two typical cases: 1) at the very beginning when the training loss decreases the fastest, synchronization can help to avoid staleness of parameters afterwards; 2) when some workers terminate their computation steps in a similar time, it is worth to spend a small amount of time to synchronize in order to reduce staleness. Therefore, RLP exhibits a lower level of staleness and needs much less iterations (only around $58\%$) than ASP. Moreover, the extra time spent on inference in RLP to generate actions only takes near $1\%$, so applying RLP adds very little extra cost to the SGD training process.

## 5 CONCLUSION AND FUTURE WORK

We have presented an RL-based framework used to learn synchronization policies for PS-based training with distributed SGD. Based on the results of our experiments we argue the following points:

- There exist synchronization policies for PS-based training beyond classic BSP, ASP and SSP which lead to shorter training time and improved resource utilization. It should be a worthwhile research endeavour to explore this topic for different types of models, in particular computationally demanding models.
- Such synchronization policies may even provide better results in terms of used computational resources when used with slightly modified models, different cluster environments and training data.
- The additional overhead incurred by having to perform an extra inference step for the RL policy network in the parameter server may still be less than the overall gain in efficiency from the policy when compared to BSP, ASP and SSP.

For future work we plan to deploy RLP in a real-world scenario on a range of different model classes and gain further insights into its practicability. Moreover, we hope to further formalize our approach in order to gain theoretical insights into the existence of optimal synchronization policies.

We further believe that reinforcement learning might be a valuable research tool for the distributed systems community supporting the exploration and discovery of new policies for control problems which are typically encountered in this field.

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
