# OpenReview forum: "Learning Efficient Parameter Server Synchronization Policies for Distributed SGD"
_ICLR.cc/2020/Conference — Accept (Poster)_

### Official Review · AnonReviewer3 · 2019-10-22
**Official Blind Review #3**

**Rating:** 6

**Review:**

This paper studies how to improve the synchronization policy for parameter server based distributed SGD algorithm. Existing work focus on either synchronous or asynchronous policy, which often results in straggler or staleness problems. Recent research proposes different ways to make the policy design fully adaptive and autonomous. This paper proposes a reinforcement learning (RL) approach and shows promising results to reduce the total training time on various scenario.

A major challenge is to design the state and action spaces in the reinforcement learning setting. It requires the design can be generalized to different scenario, while ensuing efficient policy learning. Compared to existing policies such as BSP, ASP and SSP, RL has an advantage to adapt to non-stationary situations over the training process. Compared to other existing approaches, RL could be fully data-driven.

The paper formalizes an RL problem by minimizing the total training time to reach a given validation accuracy. To minimize the number of actions, only 3 actions coming from BSP, ASP and SSP are used. The state space is designed to capture similar loss curves, and to be independent of the number of workers (as much as possible). A policy network is used make decisions after trained with exiting methods.

Numerical results validate that the RL policy improves the training time compared to BSP, ASP and SSP. Different number of works and models, dataset are also tested to show the RL policy is generalizable to unseen scenario. Although all the results are simulated in a controlled environment, Figure 4 gives a very interesting illustration showing the advantage of using the RL policy. I still have detailed comments (see below), but I find the paper well written, and the author(s) has obtained promising results.

Detailed comments:
-	The validation accuracy 88% on MINST seems pretty low to me to stop the algorithm, in particular when training multiple layer neural networks. What would happen if the accuracy is increased, can the RL approach still find a good policy? What about the validation accuracy on CIFAR-10?
-	I still have some concern of the computation time obtain the RL state per step. In particular, the time cost to compute the loss L on different weights w. How do you address this issue? Is L computed on the validation set? What is its size? This parameter seems to me highly sensitive when the policy is used for different dataset, in particular the dataset vary. It would be better to have more discussions in the paper or appendix.
-	What is the final test accuracy on the trained models using different policies? This allows us to see whether the approach has not over-fitted to the training/validation set.

Some typo:
Page 2 line 2 AP -> ASP
Page 5, last line 4: converge -> convergence


**Experience Assessment:**

I have published one or two papers in this area.

**Review Assessment: Checking Correctness Of Derivations And Theory:**

N/A

**Review Assessment: Checking Correctness Of Experiments:**

I assessed the sensibility of the experiments.

**Review Assessment: Thoroughness In Paper Reading:**

I read the paper at least twice and used my best judgement in assessing the paper.

---

> ### Author Response · Authors · 2019-11-11
> **Response to Reviewer 3 (Problems VII and VIII)**
>
>  In the following, we list your concerns on the Problems VII and VIII and our detailed responses.
>
> Problem VII:
> What is the final test accuracy on the trained models using different policies? This allows us to see whether the approach has not over-fitted to the training/validation set.
>
> Response to Problem VII:
>
> Thank you for your reminder. We put this data in the Table 2, Appendix A, Page 12, in the revised paper. We report the test accuracy on the DNN models trained by different policies (BSP, ASP, SSP and RLP). Meanwhile, according to comments, we also report the test accuracy of the models trained with different validation accuracy bounds (88%, 92% and 95%). We have three observations:
>
> 1) the final test accuracy is close to the validation accuracy bound, which indicates the approach does not overfit the training/validation set.
>
> 2) the final test accuracy of BSP is a bit higher than others. This is because there is no staleness effect in BSP so it is likely to be more accurate. However, BSP is the slowest policy of all of them.
>
> 3) the final test accuracy of RLP is a bit higher than ASP. This is simply because there is no synchronization barriers in ASP while RLP does some synchronizations adaptively to reduce the staleness effects.
>
> ---------------------------------------------------------------------------------------------------------------------------------------------------------
>
> Problem VIII:
> Some typo:
> Page 2 line 2 AP -> ASP
> Page 5, last line 4: converge -> convergence
>
> Response to Problem VIII:
> Thank you very much for your careful review. We have fixed the typos and proofread the paper several times.

---

> ### Author Response · Authors · 2019-11-11
> **Response to Reviewer 3 (Problem VI)**
>
> In the following, we list your concerns on the Problem VI and our detailed responses.''
>
> Problem VI:
> I still have some concern of the computation time obtain the RL state per step. In particular, the time cost to compute the loss L on different weights w. How do you address this issue? Is L computed on the validation set? What is its size? This parameter seems to me highly sensitive when the policy is used for different dataset, in particular the dataset vary. It would be better to have more discussions in the paper or appendix.
>
> Response to Problem VI:
> Thank you for your kind review. At first, let us present more details on how to compute the RL state per iteration. As we show in Section 3.2, Page 4, for each iteration n, we record the feature tuple $T_n = (n, L(w), L(w'), l_n)$ and the state vector $S_n$ keeps track of the H most recent feature tuples.
>
> In the tuple $T_n$, L(w) represents the loss value of the underlying trained ML model computed by the worker X which submits the gradient in the n-th iteration. It is computed on the worker side but not the PS side. The argument w represent the model parameter previously pulled and stored locally by the worker X. It fetches a mini-batch of the training data, obtains the loss L(w) by the forward propagation and the gradient by the backward propagation. L(w) and the gradient value are then submitted together to the PS side.
>
> L(w') represents the loss value of the underlying trained ML model computed by the worker Y submitting the gradient in the previous (n-1)-th iteration. It is computed by Y in the same way and stored in the PS side. Anytime when a new worker finishes and submits a new loss value, we can discard the previous loss value L(w'), set the current loss value L(w) to be the previous one and the new loss value to be the current one.
>
> Therefore, there is no computation cost on the PS side to get L(w). We directly apply the loss computed by workers on the training data to generate the state vector.
>
> As for the sensitivity of loss value w.r.t. different dataset, we admit that until now we only train and evaluate on models and datasets with similar loss curves. We put into our future work one  important direction to improve the generality of the RL policy to different models and datasets. We are now working along two directions: 1) designing  methods to normalize the loss value; and 2) train the RL policy on more models with different loss curves. Nevertheless, we argue that our current results clearly indicate the applicability of our RL-based approach in practice.

---

> ### Author Response · Authors · 2019-11-11
> **Response to Reviewer 3 (Problem V)**
>
> In the following, we list your concerns on the Problem V and our detailed responses.
>
> Problem V:
> The validation accuracy 88% on MINST seems pretty low to me to stop the algorithm, in particular when training multiple layer neural networks. What would happen if the accuracy is increased, can the RL approach still find a good policy? What about the validation accuracy on CIFAR-10?
>
> Response to Problem V:
> Thank you for your kind review. In our experimental settings, we set the validation accuracy bound to 88% as the termination condition for each testing case on the MNIST dataset. We elaborate the two reasons for choosing such value bound as follows.
>
> On one hand, taking a relative lower bound saves  evaluation time and allows us to quickly examine the performance of our proposed method. We observed that the accuracy of the underlying trained DNN models improves in general much slower when the accuracy value is high. Therefore, for large accuracy bounds, each training instance needs more time to terminate. In our experiments, each testing case involves running thousands of instances to train the RL policy, so we just take a relative lower accuracy bound. In  Ref.[Wei Dai et.al., ICLR'19], a benchmark evaluation on the staleness effects,  a lower validation accuracy bound (71%) is also adopted for fast examination.
>
> On the other hand, we argue that the experimental results obtained on 88% are representative enough to exhibit the superiority of RLP. To avoid confusion, we do more testing and provide the experimental results with higher validation accuracy bounds, i.e., 92% and 95%, in the Appendix A, Page 11. From Figure 5 in Page 11, we find that RLP still runs much faster than BSP, ASP and SSP on higher validation accuracy bounds. The only difference is that the speedup ratio of RLP w.r.t. BSP (and SSP) increases for higher accuracy bound while the speedup ratio of RLP w.r.t. ASP decreases for higher accuracy bound. We explain the reasons in Appendix A, Page 11. In the training stage where the underlying trained models attain a high accuracy,  the gain of each iteration is already very small, so the staleness effect is not significant at this time. All synchronization policies will have to perform a large number of iterations to converge. Therefore, the best synchronization policy for this training stage is ASP, which costs less time that BSP and SSP. In our case, RLP is able to learn to act as ASP in this training stage. Observing Figure 4(d), Page 8, we find that RLP has very less synchronization barriers and acts very similar to ASP near the end of the training process. As a result, for higher validation accuracy bound, the time difference widens between RLP and BSP/SSP while narrows between RLP and ASP.
>
> For the CIFAR-10 case, our underlying DNN model has a relatively small capacity and hence cannot attain high accuracy on this more complex dataset. We thus set the validation accuracy bound to 35% in the generalization experiments. We have added into as a footnote in the paper.

---

### Official Review · AnonReviewer2 · 2019-10-23
**Official Blind Review #2**

**Rating:** 3

**Review:**

This paper proposes to use deep RL to learn a policy for communication in the parameter-server setup of distributed training. From the perspective, the problem formulation is a nice contribution.

While it is a reasonable idea and the initial results are promising, the lack of an evaluation on a real cluster, or for training more computationally-demanding models, is limiting. I fully appreciate the need to perform experiments in a controlled environment, such as the ones reported in the paper. These are useful to validate the idea and explore its potential limitations. However, to truly validate such an idea completely it is also necessary to implement it and run it "in the wild" on an actual distributed system. From my experience, although performing such experiments is certainly more involved and challenging, there can also be significant differences in the outcomes when one goes to such an implementation. Normally these are due to discrepancies between the assumed/simulated model, and real system behavior.

Is it clear that deep RL is needed for this application, as opposed to more traditional RL approaches (either tabular, with suitably quantized actions, or a simpler form of function approximation? And to ask in the other direction, did you consider using a more complex policy architecture, e.g., involving an LSTM or other recurrent unit?



**Experience Assessment:**

I have read many papers in this area.

**Review Assessment: Checking Correctness Of Derivations And Theory:**

N/A

**Review Assessment: Checking Correctness Of Experiments:**

I carefully checked the experiments.

**Review Assessment: Thoroughness In Paper Reading:**

I read the paper at least twice and used my best judgement in assessing the paper.

---

> ### Author Response · Authors · 2019-11-11
> **Response to Reviewer 2 (Problem IV)**
>
> In the following, we list your concerns on the Problem IV and our detailed responses.
>
> Problem IV:
> Is it clear that deep RL is needed for this application, as opposed to more traditional RL approaches (either tabular, with suitably quantized actions, or a simpler form of function approximation? And to ask in the other direction, did you consider using a more complex policy architecture, e.g., involving an LSTM or other recurrent unit?
>
> Response to Problem IV:
> Thank for your helpful suggestions. In general, the key challenge for our problem is to design the state and action space, while still balancing cost and benefit. We believe that in our case the design of a deep but not very complex network and a small but powerful action space allows us to get the most benefits out of an RL-based approach, while preserving both training and inference efficiency. The details are as follows.
>
> On one side, in terms of benefits, the state space must be able to generalize to different scenarios, such as a different number of workers, datasets and underlying models to be trained. Therefore, we choose some features in the state to characterize the execution process and they are independent of the number of workers, datasets and underlying models (See the paragraphs in Page 4 entitled with "State" for more details). The experimental results in Section 4 have verified the generality of our state design (See the paragraphs in Page 7 entitled with "Generality of RLP" and Figure 3 for more details). The state space in our problem is very complex and large and may contain dozens to hundreds of features. In our experimental setting, it contains 40 dimensions and some dimensions are continuous (such as the loss values). Therefore, we use a deep neural network to represent the transition function in RL and apply DQN to train the RL policy. The traditional tabular algorithms or simpler form of function approximations have difficulties representing the large and complex transition function in this application.
> Regarding the action space, it is discrete. Therefore, there is no need to quantize the action space. It would, when designed naively, contain up to 2^n actions for n workers since for each worker it need to be decided whether to run or wait, respectively. For efficiency reasons, we thus choose a small action space with only three valid actions. However, it is powerful enough to generalize all existing policies (See the paragraphs in Page 5 entitled with "Action" for more details).
>
> On the other side in terms of cost, the design must ensure efficient policy learning and action inference. At the very beginning, we have considered to use LSTM or other recurrent units. Then we have chosen a different approach for two reasons: 1) LSTM-based architectures tended to diverge during training. A possible reason may be the non-stationary nature of our RL problem, so more complex models with sequential information have convergence issues. 2) An LSTM-based architecture would lead to higher computational costs for the action inference. Using our method, the inference time only takes around 1% when training a model using our RL-based policy. Therefore, we eventually decided to use a simple two-layer neural network in our setting to approximate the RL transition function.
> To capture the sequential information of the training steps in our model, we instead encode them in our state representation. That is, each state not only contains the information of the current step, but also contains the historical information of some previous steps.
>
> Regarding our choices we were inspired by the following two papers.
> [a] Andrychowicz, Marcin, et al. "Learning to learn by gradient descent by gradient descent." NIPS. 2016.
> [b] Li, Ke and Jitendra, Malik. "Learn to optimize." ICLR. 2016.
>
> Paper [a] applies the LSTM with supervised learning while paper [b] encodes historical information into state with reinforcement learning. As claimed by authors of [b]
> (Available on https://bair.berkeley.edu/blog/2017/09/12/learning-to-optimize-with-rl/  ),
> the method in [a] is hard to generalize to unseen cases and may diverge while the method in [b] is more general and easy to train.

---

> > ### Comment · AnonReviewer2 · 2019-11-15
> > **Acknowledging responses**
> >
> > Thank you for your responses to my review. I appreciate the effort that went into running the initial experiments on a real cluster. At this stage, I still don't feel that this evaluation is sufficient to convince me to raise my review score. I would expect that the workload (e.g., larger dataset and model) can have a significant effect on the performance of such an approach and further investigation is needed to obtain more convincing results.

---

> ### Author Response · Authors · 2019-11-11
> **Response to Reviewer 2 (Problem III)**
>
> In the following, we list your concerns on the Problem III and our detailed responses.
>
> Problem III:
> While it is a reasonable idea and the initial results are promising, the lack of an evaluation on a real cluster, or for training more computationally-demanding models, is limiting. ... (omitted due to space limit) ... Normally these are due to discrepancies between the assumed/simulated model, and real system behavior.
>
> Response to Problem III:
> Thank you very much for your very helpful reviews and suggestions. We very much appreciate your understanding of the challenges of the experiments in a real distributed system. In fact, we have been following a three-step plan for this work which almost follows your recommendation.
>
> For the first step, we have implement a simulated environment to validate the idea and explore  potential limitations. In the second step, we have implemented a prototype distributed system to further examine the method and address possible problems. In the third step, we plan to implement this method as an builtin component for Tensorflow, so that it may be used in real-world production scenarios.
>
> This papers mainly shows our results for the first step and concentrates on exhibiting our novel ideas in applying RL into synchronization policy design. After submission, we have implemented a prototype distributed system and further tested our method. We build the system by directly following the standard PS architecture which contains one PS node and multiple worker nodes. They are connected through a network. Based on this real distributed system, we do more experiments and present you the outline as follows. We have added the new experimental results in the Appendix B, Pages 12-13, in the revised paper.
>
> First, for the experimental setting, we use the same settings for  the hyperparameters as in the  experiments in the simulated environment. We use the same method to generate instances to train the RL policy except that we do not use the sleep() function to simulate the behavior of stragglers. Before training, we also apply existing policies to pre-train the policy network of RL. This time, we find that RLP needs to be trained with more instances (around 1, 500 to 2000 episodes) to converge, since the situation is more complex.
>
> Second, for comparing the performance of RLP w.r.t. existing policies, we find that RLP still outperforms them, and sometimes performs even better than the simulation environment. As shown in Figure 6, RLP is 2.11 and 1.64 times faster than BSP and SSP, respectively, which are higher than the simulation environment; and RLP is 1.28 times faster than ASP, which is lower than the simulation environment. We conjecture that the key reason underlying this is that the straggler effect is much more significant than the staleness effect in a real cluster environment. In real clusters, the variance of running times of different workers tends to be higher, hence more workers are likely to be stragglers. Meanwhile, the network may delay the updates of some workers, which further amplify the straggler effects. However, the staleness effect is a property which is more closely related with the trained ML model itself and not as vulnerable as the straggler effect due to the cluster environment. Therefore, ASP runs faster than BSP and SSP on real clusters. As a result, our RLP improves more on BSP and SSP while less on ASP. Notice that, this result once again verifies the adaptivity of our RLP method, which can find better policies both in the simulation environment and real cluster environments.
>
> Third, on the generality of the RLP, as shown in Figure 7, we observe that the trained RLP policy can generalize to clusters with different numbers of workers, new models and new data. This is because in our RL formulation, both the state and action representation are irrelevant to the number of workers. Meanwhile, we record only the loss value information  in the state information of RLP. Thus, training models with similar loss curve may also speed up by our RLP policy experience.
>
> In this paper, we mention in limitations (Page 2) that we do not apply our method on very demanding models such as CNN or BERT-like model due to resource constraints. We are currently implementing RLP in order to deal with these computationally-demanding models more efficiently in a  distributed environment. However, these models are much more complex and needs more time to train and tune, so the results are not yet ready to be provided in this paper. We will address this in future work.
>
> Regarding an integration into Tensorflow we are currently in discussion regarding a modification of the token queue mechanism underlying TensorFlow's builtin PS. Ultimately, we  want to integrate RLP into the generation and fetching procedures of the token queue in order to implement our synchronization policy beyond the prototype distributed implementation.

---

### Official Review · AnonReviewer1 · 2019-10-24
**Official Blind Review #1**

**Rating:** 6

**Review:**

Background disclaimer: I work in RL research for quite an amount of time, but I do not know much about the domain of distributed systems. For this reason, I may not know the details of technical terms, and I might not be the best person to review this work (when compared with the literature in this field). Nevertheless, below I try to give my evaluation based on reading the paper.

====================
In this work, the authors applied value-based reinforcement learning to learn an optimal policy for global parameter tuning in the parameter server (PS) that trains machine learning models in a distributed way using  stochastic gradient descent. Example parameters include SGD hyper-parameters (such as learning rate) and system-level parameters. Immediate cost is to minimize the training time of the SGD algorithm, and i believe the states are the server/worker parameters. From the RL perspective, the algorithm used here is a standard DQN with discrete actions (choices of parameters). But in general I am puzzled why the action space is discrete instead of continuous, if the actions are the hyper-parameters. State transition wise, I am not sure if the states follow an action-dependent MDP transition, and therefore at this point I am not sure if DQN is the best algorithm for this applications (versus bandits/combinatorial bandits).  While it is impressive to see that RL beats many of the SOTA baselines for parameter tuning, I also find that instead of using data in the real system to do RL training, the paper proposes generating "simulation" data by training a separate DNN. I wonder how the performance would differ if the RL policy is trained on the batch real data.



**Experience Assessment:**

I do not know much about this area.

**Review Assessment: Checking Correctness Of Derivations And Theory:**

I assessed the sensibility of the derivations and theory.

**Review Assessment: Checking Correctness Of Experiments:**

I assessed the sensibility of the experiments.

**Review Assessment: Thoroughness In Paper Reading:**

I made a quick assessment of this paper.

---

> ### Author Response · Authors · 2019-11-11
> **Response to Reviewer 1 (Problem II)**
>
> In the following, we list your concerns on the Problem II and our detailed responses.
>
> Problem II:
> While it is impressive to see that RL beats many of the SOTA baselines for parameter tuning, I also find that instead of using data in the real system to do RL training, the paper proposes generating "simulation" data by training a separate DNN. I wonder how the performance would differ if the RL policy is trained on the batch real data.
>
> Response to Problem II:
> Thank you very much for your very helpful suggestions. In this paper, we mainly concentrated on exhibiting our novel ideas for applying RL to synchronization policy design. Therefore, we implemented a simulated environment which is much easier to configure and to validate the idea and explore the potential limitations. However, this is only the first step of our work. In the second step, we have implemented a prototype distributed system to further examine the method and fix possible problems. In the third step, we plan to implement this method as an component for Tensorflow, so that it may be used in real-world production scenarios.
>
> After the submission, we have worked on the second step and implemented a prototype distributed system and further tested our method. This system runs on a real cluster environment, the implementation uses the Tensorflow framework. Based on this real distributed system, we have done more experiments. We have added the new experimental results in the Appendix B, Pages 12-13, in the revised paper. Some outlines on the results are presented as follows.
>
> First, RLP still outperforms existing policies BSP, ASP and SSP on the real cluster. As shown in Figure 6, RLP is 2.11 and 1.64 times faster than BSP and SSP, respectively, which are higher than the simulation environment; and RLP is 1.28 times faster than ASP, which is lower than the simulation environment. We conjecture that the key reason underlying this phenomenon is that the straggler effect is more significant than the staleness effect in the real cluster environment. Therefore, ASP runs much faster than BSP and SSP (almost 1.5 to 2 times) on real clusters, and our RLP improves more on BSP and SSP and less on ASP. Notice that this result once again verifies the adaptivity of our RLP method, which can find better policies both in the simulation environment and real cluster environment with different levels of straggler and staleness effects.
>
> Second, on the generality of the RLP, as shown in Figure 7, we observe that the trained RLP policy can generalize to clusters with different number of workers, new models and new data. This is due to the fact that in our RL formulation, both the state and action representation have no connection to the number of workers. Therefore RLP can transfer to different number of workers. Meanwhile, we record only the loss values in the state information of RLP. Thus, training models with similar loss curve may also incur a speed up by our RLP policy. Once again these results verify that our state and action design for RLP are reasonable and effective.
>
> In this, we do not apply our method on very demanding models such as CNN or BERT-like models due to resource constraints. We are currently implementing RLP in order to deal with these computationally-demanding models more efficiently in a distributed environment. However, these models are much more complex and needs more time to train and tune, so the results are not yet ready to be provided in this paper. We will address this in future work.
>
> Regarding an integration into Tensorflow we are currently in discussion regarding a modification of the token queue mechanism underlying TensorFlow's builtin PS. Ultimately, we want to integrate RLP into the generation and fetching procedures of the token queue in order to implement our synchronization policy beyond the prototype distributed implementation.

---

> ### Author Response · Authors · 2019-11-11
> **Response to Reviewer 1 (Problem I)**
>
> In the following, we list your concerns on the Problem I and our detailed responses.
>
> Problem I:
> In this work, the authors applied value-based reinforcement learning to learn an optimal policy for global parameter tuning in the parameter server (PS) that trains machine learning models in a distributed way using  stochastic gradient descent. Example parameters include SGD hyper-parameters (such as learning rate) and system-level parameters. Immediate cost is to minimize the training time of the SGD algorithm, and i believe the states are the server/worker parameters. From the RL perspective, the algorithm used here is a standard DQN with discrete actions (choices of parameters). In general I am puzzled why the action space is discrete instead of continuous, if the actions are the hyper-parameters. State transition wise, I am not sure if the states follow an action-dependent MDP transition, and therefore at this point I am not sure if DQN is the best algorithm for this applications (versus bandits/combinatorial bandits).
>
> Response to Problem I:
> Thank you for your kind review. First let us clarify the difference of our work with learning hyperparameters, then answer the question on actions and states.
>
> In this paper, we learn an optimal ''synchronization policy'' used for the distributed training of machine learning models with Stochastic Gradient Descent (SGD) in Parameter-Server (PS)-based environment. This setting consists of one (or several) PS maintaining model parameters and receiving updated gradients from workers, and multiple workers pulling model parameters from PS, computing gradients and pushing them back to PS.
>
> The synchronization policy is a mechanism to coordinate the execution progress of all workers in the PS setting. It determines in each step, i.e., whenever a worker pushes its gradient to the PS, whether this worker should continue to run for the next step or wait for sometime for the completion of some other workers. Thus, it is independent of the hyper-parameters of SGD and system parameters. Hence, we are trying to optimize the mechanism of the synchronization policy to save training time but not tuning the global hyperparameters of SGD.  In short, our work falls in the category of "learning how to learn" to train the underlying ML models while hyperparameter tuning falls in the category of "learning which model to learn" to optimize the hyperparameters of the underlying ML models.
>
> To  this end, we formalize the design of a synchronization policy as a reinforcement learning problem (see Figure 1 in Page 5 for an illustration). In this RL problem, for the state we choose  features which characterize the execution progress of SGD training in each step. To ensure the expression power of the state, the state space in our problem is large when compared to more standard RL problem instances. Each state vector may contain dozens to hundreds of features (See the paragraphs in Page 4 entitled with "State" for more details). Therefore, we choose a deep neural network to represent the transition function \pi(S, a) and apply DQN to train the RL policy. The tabular and bandits algorithms are unable to represent the large and complex transition function in this application.
>
> In our RL problem, each action represents a decision for each worker to run or wait at each step. Therefore, the action space is discrete. It contains at most 2^n actions for n workers since for each worker it need to be decided whether to run or wait, respectively. We choose a small but powerful action space containing three valid actions to enable fast training of the RL policy (See the paragraphs in Page 5 entitled with "Action" for more details). We design the state and action in this manner in our RL setting to ensure its generality while keeping training and inference efficiency.
>
> The state clearly follows an action-dependent MDP transition since the next position of execution process is purely determined by the current state (where we are) and the next action (where we go).

---

### Author Response · Authors · 2019-11-11
**Responses to all Reviewers**

We would like to thank all reviewers for their kind reviews and helpful suggestions.

According to the reviews, we summarized all questions into 8 problems. To clarify some aspects, we write detailed responses to each problem in the individual comment to each reviewer. Meanwhile, we have substantially revised the paper to provide  more experimental results in the Appendix, Pages 11-13. In the following, we present a short summary of points raised in the reviews and our responses. Please refer to the individual comments and the revised paper for more details.

=======================================    Summary     ==========================================
--> For Reviewer 1:
Problem I: Possible confusions of our work with hyperparameter learning, the state/action space and the reasons for using DQN.

Response I: We clarify the difference of our work with learning hyperparameters: our work falls in the category of "learning how to learn" while hyperparameter tuning falls in the category of "learning which model to learn". In our RL  formulation, each action represents a decision to control whether each worker should run or wait, thus it is discrete. The state clearly follows an action-dependent MDP transition. We choose a DQN since the simple tabular method cannot process the large and complex transition function in this application.

Problem II: Experimental results on real system data.

Response II: We have provided more experimental results obtained from a prototype implementation on a real distributed cluster environment in Appendix B, Pages 12-13, in the revised paper. The results show that our method still outperforms existing policies in a real system. Meanwhile, our method can generalize to unseen cases in a real system. We also explain the reasons causing the differences between the results for the simulated and the real environment.


--> For Reviewer 2:
Problem III: Experimental results on applying RL in real clusters.

Response III:  We have provided more experimental results obtained from a prototype implementation on a real distributed cluster environment in Appendix B, Pages 12-13, in the revised paper. The results show that our method still outperforms existing policies in a real system. Meanwhile, our method can generalize to unseen cases in a real system. We also explain the reasons causing the differences between the results for the simulated and the real environment. Based on these difference we can verify the adaptability of our method. Moreover, we introduce our plan for future work.

Problem IV: The reasons for choosing deep RL but not other simple or complex models.

Response IV: Simpler models cannot represent our transition function while  complex models would be too costly to train and use for inference during the actual training. We clarify that in our application the design of a deep but not very complex network and a small but powerful action space allows us to get the most benefits out of an RL-based approach, while preserving both training and inference efficiency.


--> For Reviewer 3:
Problem V: Performance of RL with other validation accuracy bound.

Response V: The main reason for choosing the validation accuracy bound as a relative low value is that this allows for faster experiment cycles. We have also provide  experimental results for RLP on higher validation accuracy bounds in Appendix A, Page 12, in the revised paper. Our method still outperforms the other policies, and we explain the reasons for the difference of results with different validation accuracy bounds.

Problem VI: Confusions on the computation of the loss.

Response VI: The loss is computed on the training data by workers and submitted to the PS side, so it incurs no extra cost.

Problem VII: Test accuracy of models trained by different policies.

Response VII: We provide the test accuracy of models trained by different policies in Appendix A, Pages 12--13, in the revised paper. The results verify that our method does not overfit the training and validation dataset.

Problem VIII: Some typo errors.

Response VIII: We have fixed the mentioned typos and have proofread the paper several times.
==============================================================================================

We ask you to refer to the individual comments per problem of per reviewer and and consider the Appendix in the revised paper for more details on our responses.

Thank you very much again for all of your hard work and careful reviewing!

---

### Decision · Program_Chairs · 2019-12-19

**Decision:**

Accept (Poster)

**Comment:**

The authors consider a parameter-server setup where the learner acts a server communicating updated weights to workers and receiving gradient updates from them. A major question then relates in the synchronisation of the gradient updates, for which couple of *fixed* heuristics exists that trade-off accuracy of updates (BSP) for speed (ASP) or even combine the two allowing workers to be at most k steps out-of-sync. Instead, the authors propose to learn a synchronisation policy using RL. The authors perform results on a simulated and real environment. Overall, the RL-based method seems to provide some improvement over the fixed protocols, however the margin between the fixed and the RL get smaller in the real clusters. This is actually the main concern raised by the reviewers as well (especially R2) -- the paper in its initial submission did not include the real cluster results, rather these were added at the rebuttal. I find this to be an interesting real-world application of RL and I think it provides an alternative environment for testing RL algorithms beyond simulated environments.   As such, I’m recommending acceptance. However, I do ask the authors to be upfront with the real cluster results and move them in the main paper.